# Genetic control of the leaf ionome in pearl millet and correlation with root and agromorphological traits

Princia Nakombo-Gbassault[1,2], Sebastian Arenas[1¤], Pablo Affortit[1], Awa Faye[3], Paulina Flis[4], Bassirou Sine[3], Daniel Moukouanga[1], Pascal Gantet[1], Ephrem Kosh Komba[2], Ndjido Kane[3], Malcolm Bennett[4], Darren Wells[4], Philippe Cubry[1], Elizabeth Bailey[4], Alexandre Grondin[1], Yves Vigouroux[1], Laurent Laplaze[1]*

1 DIADE, Université de Montpellier, IRD, CIRAD, Montpellier France, 2 JEAI AgrobiodiveRCA, Université de Bangui, Bangui, Central African Republic, 3 CERAAS, Institut Sénégalais des Recherches Agricoles (ISRA), Thiès, Senegal, 4 School of Biosciences, University of Nottingham, Sutton Bonington, United Kingdom

¤Current address: Université Côte d'Azur, CNRS, INSERM, Institute for Research on Cancer and Aging (IRCAN), Nice, France
* laurent.laplaze@ird.fr

## Abstract

Pearl millet (*Pennisetum glaucum*) thrives in arid and nutrient-poor environments, establishing its role as a crucial cereal crop for food security in sub-Saharan Africa. Despite its remarkable adaptability, its yields remain below genetic potential, primarily due to limited water and nutrient availability. In this study, we conducted ionomic profiling and genome-wide association studies (GWAS) in field conditions across two growing seasons to unravel the genetic basis of nutrient acquisition in pearl millet. Soil ion content analyses revealed significant differences in nutrient distribution between field sites, while certain ions, such as phosphorus (P) and zinc (Zn), consistently displayed stratified accumulation patterns across years, suggesting stable depth-dependent trends. Evaluation of a genetically diverse panel of inbred lines revealed substantial variation in leaf ion concentrations, with high heritability estimates. Correlations between leaf ion content and root anatomical or agromorphological traits highlighted the intricate interplay between genetic and environmental factors shaping leaf ion accumulation. These analyses also uncovered potential trade-offs in nutrient acquisition strategies. GWAS identified genomic regions associated with leaf ion concentrations, and the integration of genetic and gene expression data facilitated the identification of candidate genes implicated in ion transport and homeostasis. Our findings provide valuable insights into the genetic regulation of nutrient acquisition in pearl millet, offering potential targets for breeding nutrient-efficient and climate-resilient varieties. This study underscores the importance of integrating genetic, physiological, and root architectural traits to enhance agricultural productivity and sustainability in resource-constrained environments.

**Data availability statement:** Plant and soil ion content data, plant agromorphological data, GBS data as well as scripts are available in the DataSuds database (https://doi.org/10.23708/DWGEAJ). RNAseq data are available from the Gene Expression Omnibus at GenBank (https://www.ncbi.nlm.nih.gov/geo/; accession number: GSE286898).

**Funding:** P.N.G. was supported by a joint PhD grant from the Institut de Recherche pour le Développement (IRD, France) and the French Embassy in the Central African Republic. We acknowledge support from the Royal Society (Anatomics grant ICA-R1-180356 to MB and 785 NK), the Agropolis Fondation (ValoRoot grant n°2202-002 to LL) as part of the "Investissement d'avenir" (ANR-I0-LABX-0001-0I), under the frame of I-SITE MUSE (ANR-16-IDEX-0006), and the European Plant Phenotyping Network (EPPN2020 project Ionomil n°386). The funders had no role in study design, data collection and analysis, decision to publish, or preparation of the manuscript.

**Competing interests:** The authors have declared that no competing interests exist.

## Introduction

Pearl millet (*Pennisetum glaucum*) is a nutrient-rich cereal crop widely cultivated in arid and semi-arid regions, particularly in sub-Saharan Africa, where it serves as a critical source of grain and fodder for millions of smallholder farmers [1,2]. Domesticated approximately 4,500 years ago in the Sahel region [3], pearl millet can yield in hot, dry climates and nutrient-poor soils, making it a strategic crop for enhancing agricultural resilience under climate change scenarios [4–7]. Despite its exceptional adaptability, pearl millet yields in sub-Saharan Africa remain low, averaging 800–1,000 kg/ha, far below its genetic potential [8]. To improve productivity and nutritional quality, it is essential to better understand the physiological and genetic mechanisms governing the uptake, accumulation, transport, and utilization of nutrients, as these processes are crucial for plant growth, development, and resilience to biotic and abiotic stresses [9].

The plant ionome, which reflects the composition of mineral nutrients and trace elements, is influenced by genetic, environmental, and developmental factors [10–12]. Optimal plant growth requires a balanced supply of macronutrients such as potassium (K), calcium (Ca), and magnesium (Mg), as well as micronutrients like iron (Fe), zinc (Zn), and copper (Cu) [10,11]. For instance, potassium improves water-use efficiency, calcium contributes to cellular signaling and stress responses, and magnesium supports chlorophyll synthesis and enzymatic activities critical for photosynthesis [13–15]. Efficient acquisition, transport, and utilization of these nutrients are particularly important in resource-constrained environments. Additionally, as pearl millet is a vital forage source in drylands, enhancing its nutrient-use efficiency is essential for improving both grain and forage quality. Conversely, plants must mitigate the uptake of toxic elements such as arsenic (As), cadmium (Cd), and lead (Pb) to prevent detrimental accumulation [16].

High-throughput elemental profiling combined with genetic analyses has significantly advanced ionomics research, with applications now extending to crops such as rice, maize, barley, soybean, and tomato [17–22]. Integrating ionomic profiling with genome-wide association studies (GWAS) offers a robust approach to identifying genes regulating the ionome [23]. This approach has successfully identified genes and quantitative trait loci (QTLs) associated with mineral accumulation, including *Heavy Metal ATPase 3* (*HMA3*) for cadmium and *molybdate transport proteins 1* (*MOT1*) for molybdenum in Arabidopsis [10,24], *high-affinity K+ transporter 1;5* (*HKT1;5*) for sodium in rice [25], and loci controlling iron, zinc, and phosphorus in maize [26].

The ionome is predominantly shaped by soil mineral availability, with roots playing a pivotal role in nutrient acquisition and transport [27,28]. The dynamic and complex architecture of root systems ensures a continuous supply of water and nutrients. Investigating the genetic correlations between root architecture and foliar ion content in pearl millet can elucidate the mechanisms underlying nutrient uptake and distribution. Identifying root traits that enhance nutrient acquisition could support the development of nutrient-efficient pearl millet varieties, thereby improving growth and yield.

The objective of this work was to identify genetic loci and root traits controlling leaf ionome in pearl millet. To address these knowledge gaps, a diverse panel of pearl millet inbred lines was analyzed for leaf ion content under irrigated field conditions across two consecutive growing seasons. Our findings provide valuable insights into the genetic determinants of nutrient acquisition and utilization, paving the way for targeted breeding programs aimed at improving pearl millet's productivity and nutritional quality.

## Materials and methods

### Plant material field trials design

A total of 175 pearl millet genotypes were studied over the two field experiments. Among those, 160 fully sequenced pearl millet inbred lines from the Pearl Millet inbred Genetic Association Panel (PMiGAP; S1 Table in S1 File) were used [29]. The PMiGAP panel comprises cultivated germplasm originating from Africa and Asia and is representative of the cultivated genetic diversity of pearl millet [1,29]. Tift23DB that was used to produce the pearl millet reference genome [1] was also included along with nine inbred lines from West Africa and five Senegalese elite lines previously described in [30].

### Field trials design and morphological phenotyping

Field trials were conducted during the 2021 and 2022 growing seasons at the Centre National de la Recherche Agronomique (CNRA) of the Institut Sénégalais des Recherches Agricoles (ISRA) in Bambey, Senegal (14.42°N, 16.28°W) as previously described [30]. Field trials were conducted in collaboration with and with the permission of the ISRA. Each trial comprised 160 genotypes, with 145 genotypes in common between both years. Separate fields within the station were used in 2021 and 2022 to avoid potential residual effects across the year. In both years, 13 soil samples were collected prior to planting at various locations across the fields to analyse soil mineral composition at four depth intervals: 0–20, 20–60, 60–100, and 100–140 cm (S1 Fig in S1 File).

At 49 days after sowing (DAS) in 2021 and 42 DAS in 2022, three representative plants per plot were harvested. Root anatomy and architecture were measured as described in [30]. The last fully ligulated leaf from the main tiller of each harvested plant was collected for ion content analysis. Leaves were washed in a 0.1% Triton X-100 solution, rinsed with deionized water, and stored in paper bags before drying at 60°C in an oven for three days. The remaining shoot biomass from these plants was collected, air-dried, and weighed to estimate plant growth.

The plants remaining within the plots were maintained under full irrigation until maturity, at which point three plants per plot were harvested for agro morphological trait assessment. Measured traits included plant height, tiller number, shoot dry biomass, days to flowering, total grain weight, and thousand-seed weight (see [30] for further details).

### Soil and leaf ion content analysis

Soil and leaf ion content was measured at the University of Nottingham using Inductively Coupled Plasma Mass Spectrometry (ICP-MS,Thermo-Fisher Scientific iCAP-Q, Thermo Fisher Scientific, Bremen, Germany). The instrument employs in-sample switching between two modes using a collision cell (i) charged with He gas with kinetic energy discrimination (KED) to remove polyatomic interferences and (ii) using $H_2$ gas as the cell gas. In-sample switching was used to measure Se in $H_2$-cell mode and all other elements were measured in He-cell mode. Peak dwell times were 100 ms for most elements with 150 scans per sample.

In soil and leaves samples, the content of 21 different ions including arsenic (As), cadmium (Cd), calcium (Ca), cobalt (Co), chromium (Cr), copper (Cu), iron (Fe), potassium (K), lithium (Li), magnesium (Mg), manganese (Mn), molybdenum (Mo), sodium (Na), nickel (Ni), phosphorus (P), lead (Pb), rubidium (Rb), selenium (Se), sulphur (S), strontium (Sr) and zinc (Zn) were determined. Internal standards for scandium (Sc, 10 µg $L^{-1}$), germanium (Ge, 10 µg $L^{-1}$), rhodium (Rh, 5 µg $L^{-1}$), and iridium (Ir, 5 µg $L^{-1}$), were used to correct for instrumental drift, and were introduced to the sample stream on

a separate line (equal flow rate) via an ASXpress unit. Calibration standards included (i) a multi-element solution with Ag, Al, As, Ba, Be, Cd, Ca, Co, Cr, Cs, Cu, Fe, K, Li, Mg, Mn, Mo, Na, Ni, P, Pb, Rb, S, Se, Sr, Ti, Tl, U, V and Zn, in the range 0–100 µg L$^{-1}$ (0, 20, 40, 100 µg L$^{-1}$) (Claritas-PPT grade CLMS-2 from SPEX Certiprep Inc., Metuchen, NJ, USA); (ii) a bespoke external multi-element calibration solution (PlasmaCAL, SCP Science, France) with Ca, Mg, Na and K in the range 0–30 mg L$^{-1}$, and (iii) a mixed phosphorus, boron and sulphur standard made in-house from salt solutions (KH$_2$PO$_4$, K$_2$SO$_4$ and H$_3$BO$_3$). The matrix of the internal standards, calibration standards and sample diluents were 2% Primar grade HNO$_3$ (Fisher Scientific, UK) with 4% methanol (to enhance ionization of some elements).

For soil ion content analysis, 5 g of dried and homogenised soil were extracted with 20 mL of 1 M NH$_4$HCO$_3$ and 5 mM diamine-triamine-penta-acetic acid (DTPA) plus 5 mL MilliQ water (18.2 MΩ.cm$^{-1}$) for 1 hour on an end-over-end shaker a low speed (150 rpm). Samples were filtered (0.22 µm) and diluted 1 in 10 with 2% HNO$_3$ prior to analysis.

For leaf ion content analysis, leaf disks were sampled from dry leaves harvested from the field at around 5 cm from the ligule. Three leaf disks (5 cm diameter) from the three plants harvested in the same plot were pooled. Leaf disks were weighed (approximately 20 mg dry tissue) and 2 mL trace metal grade nitric acid Primar Plus and 1 mL 30% H$_2$O$_2$ were added before a pre-digestion step for ~20 hours at room temperature. Samples were then digested at 115 ˚C for four hours and cooled before diluting to 10 mL with MilliQ water (18.2 MΩ.cm$^{-1}$) prior to analysis.

## Genotypic data

Genotyping-by-Sequencing (tGBS®) technology was performed using leaf samples by the Freedom Markers company (USA) on 165 genotypes (the 160 genotypes from the PMiGAP, four inbred lines from West Africa and Tif23DB). GBS libraries were prepared using the restriction enzyme Bsp1286I, pooled in 96-plex, and sequenced on the Illumina HiSeq X platform (San Diego, USA). Raw sequence reads were processed by removing adapter sequences using Cutadapt v1.8 and filtering low-quality reads (minimum mean quality = 30 and minimum length = 35 bp) using the Filter_Fastq_On_Mean_Quality.pl script from the SouthGreenPlatform [31]. Filtered reads were aligned to the *Cenchrus americanus* ASM217483v2 reference genome https://www.ncbi.nlm.nih.gov/assembly/GCA_002174835.2 using the Burrows-Wheeler Aligner (BWA) v0.7.4 [32,33]. Post-alignment processing included filtering unmapped, non-uniquely mapped, and abnormally paired reads using SAMtools v0.1.18., Picard-tools-1.119 and Genome Analysis ToolKit (GATKv3.6 algorithms IndelRealigner, UnifiedGenotyper and VariantFiltration) to identify single nucleotide polymorphisms (SNPs) [34–36]. SNPs were filtered out for missing data (≥ 50%) and minor allele frequency (< 5%) using VCFtools v0.1.13 [37], which resulted in a final set of 269,848 SNPs. Individual samples with < 80% genotyping success were also excluded.

Inference of missing data was done by first inferring population structure. A cross-entropy criterion was employed to estimate the number of ancestral groups (K varying from 1 to 10) using the sparse nonnegative matrix factorization (sNMF), and 10 replications for each K [38–41]. Based on the lowest cross-entropy criterion analysis, K was set to infer four clusters [42]. Missing genotypes were imputed using a matrix factorization approach implemented in the R LEA package v2.068 [43]. The "impute" function was applied, leveraging ancestry coefficients estimated from sNMF [41]. Subsequently, the heterozygosity rate per marker was assessed, and markers with a frequency exceeding 0.12 (95th percentile) were excluded. Finally, 254,765 SNPs from 147 genotypes were used for GWAS analysis.

## Genome-wide association study and linkage disequilibrium analyses

Genome-Wide Association Study (GWAS) was performed using four mixed models: Latent Factor Mixed Models (LFMM, [44]), Efficient Mixed-Model Association (EMMA, [45]), Bayesian information and Linkage-disequilibrium Iteratively Nested Keyway (BLINK, [46]), and Compressed Mixed Linear Model (CMLM, [47]). LFMM adjusts for population structure by using latent factors to model the unobserved genetic differences among individuals. EMMA also corrects for population structure but using an estimated kinship matrix to approximate genetic relatedness. BLINK employs a Bayesian approach to model associations between SNPs and traits, accounting for linkage disequilibrium. CMLM clusters individuals to

reduce computational complexity while maintaining effective correction for population structure. GWAS analyses were initially conducted separately for each year using Best Linear Unbiased Estimators (BLUEs), and p-values for each method were then combined across two years independently using the Fisher's combining method [48]. False Discovery Rate (FDR) estimation was performed for each trait to correct for multiple testing. To calculate a p-value threshold based on the number of independent SNPs, a pruning process was implemented with Plink v1.9 [49] to exclude highly correlated SNPs. Briefly, correlations between SNPs were calculated in an interval of 50 SNPs, with a step of 5, and SNPs with a correlation greater than 0.5 were excluded. Based on this set of independent markers, a genome-wide significance threshold corresponding to a 5% FDR was defined. Only SNPs identified by at least two GWAS methods and further supported by Fisher's combining method were retained for downstream analysis. This multi-method, cross-year validation strategy aimed to prioritize robust and stable associations.

To delimit associated genomic regions, linkage disequilibrium (LD) among all significant SNPs was evaluated, and LD blocks were defined. LD among SNPs was assessed using the squared correlation coefficient ($r^2$) with the plink software. Markers with an $r^2$ exceeding 0.75 were considered linked. Then, a 25 kb region was added on each side of correlated significant SNPs [50]. The region surrounding significant associations was used to screen for potential candidate genes, based on the functional annotation of the reference genome (ASM217483v2). Quantile-Quantile and Manhattan plots illustrating the results of GWAS were produced using the qqman package v 0.1.9 [51].

## Statistical analyses

For soil ion content, the Interquartile Range (IQR) method was employed to detect and eliminate outliers at different soil depths [52,53]. Observations falling below the first quartile or above the third quartile were identified as outliers and subsequently removed from the dataset.

The leaf ions dataset was analysed using the statgenSTA package [54]. Models were fitted using functions from SpATS version 1.0-18, considering a resolvable incomplete block design [55]. Outliers exceeding a dataset-size-based limit were removed after model fitting, and the process was repeated iteratively until no additional outliers were found. Genotypes with fewer than one replication were excluded. In the model, replication was treated as a fixed effect, while the interaction "replication x block" and genotypes were treated as a fixed effect to calculate the Best Linear Unbiased Estimator (BLUEs). BLUEs and their standard deviations for each genotype were generated, along with genetic and residual variance components, and heritability estimates. Heritability is extracted from the model when genotype was fitted as a random effect and the calculation is based on the generalised heritability formula explained in [56]. The model can be expressed as:

$$Trait = repId + repId : subBlock + genotype + \epsilon$$

where:

- repId is the fixed effect of replication

- repId:subBlock is the fixed interaction between replication and block

- genotype is the fixed or random effect of genotype

The BLUEs of ion content in the leaves were assessed for normality using the Shapiro test [57], with a 5% Bonferroni corrected threshold (p-value = 0.0023, equivalent to 0.05/17). Traits that significantly deviated from normality were subjected to a Box-Cox transformation [58,59] to achieve normality.

Means, standard deviations, and coefficients of variation were calculated for all traits. To identify the phenotypic traits explaining the most variation, Principal Component Analysis (PCA) and a non-parametric one-way Analysis of Variance

(ANOVA, [60]) were performed on the BLUEs, both within each year and between years. The data were first analyzed separately for each year. Then, the datasets from both years were combined to assess the effect of the year. ANOVA assumptions, including the homogeneity of error variances across years, were tested before conducting the combined analysis. Furthermore, Pearson correlation analyses with a confidence level of 0.95 were conducted using R for all pair-wise combinations of phenotypic traits collected in the study.

Similar analyses were performed on soil ion content to investigate the impact of soil horizons and environmental factors (year, location) on ion distribution in the soil. All statistical analyses were carried out using R version 4.3.3.

### RNA-sequencing

The pearl millet reference genotype Tift23DB was used for RNAseq experiments. Plants were grown in 400 mm x 700 mm x 20 mm rhizotrons as in [61]. After fifteen days of growth, the plexiglass was carefully removed to avoid damaging the roots and samples were taken from different root types: the primary root tip (5 cm apex), the crown root tips (5 cm apex) and lateral roots on primary roots. Roots from three plants were pooled together to make one replicate. Roots tissues from three replicates were stored at -80°C before RNA extraction using the RNAeasy Plant Mini Kit (QIAGEN, Germany). RNA sequencing was performed by the Novogene Company Limited (United Kingdom) on an Illumina platform as previously described [62].

The toolbox for generic next generation sequencing (NGS) analyses (TOGGLE, version 3; [63]) was used to obtain read counts. The TOGGLE analysis pipeline included an initial cleaning step using CutAdapt (version 3.1; [64]) before mapping the reads against the pearl millet genome (ASM217483v2; [1]) using Hisat2 (version 2.0.1; [65]). Transcriptome assembly was performed using Stringtie (version 1.3.4; [66]) with guidance from the pearl millet genes coding sequences (ASM217483v2; [1]). The percentage of mapping was checked using Samtools (version 1.9; [34]).

Analysis of the data was performed using the DIANE software (version 1.0.6; [67]). Data were normalised and differentially expressed genes were identified using the DESeq2 method (implemented in the Bioconductor package; [68]) with a minimal gene count sum across samples set at 90. The adjusted $p$-value (false discovery rate; FDR) threshold for detection of genes differentially expressed was set 0.01, and an absolute Log Fold change threshold of 1 was considered.

## Results

### Soil ion content analysis across depths, locations, and years

The mineral content of 19 elements (As, Ca, Co, Cr, Cu, Fe, K, Li, Mg, Mn, Mo, Na, Ni, P, Pb, Rb, Se, Sr, Zn), was analysed at 13 positions both in 2021 and 2022 just before the trials were set. On average, the soil at the experimental sites exhibited high concentrations of calcium (512.58 mg/kg), magnesium (122.6 mg/kg), and sodium (73.4 mg/kg), while molybdenum (0.04 mg/kg) and lithium (0.003 mg/kg) were present in low concentrations across both years. The coefficients of variation (CV) highlighted substantial spatial variability for several ions, especially for phosphorus (P), zinc (Zn), and lithium (Li), which all exhibited CVs above 80% in at least one of the two years (Table 1).

The ion content varied significantly with soil depth and year (Fig 1A). The first two principal components (Dim1: 36%, Dim2: 23%) together explained 59% of the total variance. The first axis separated topsoil-associated ions from deeper-layer ions. This gradient was consistent across years, indicating stable geochemical patterns along the soil profile. The second axis showed separation between years. Depth-dependent variations were observed for most ions. However, copper (Cu) and potassium (K) in 2021, and iron (Fe) and cobalt (Co) in 2022, did not exhibit significant stratification (Fig 1B). Shallow soil horizons contained higher levels of phosphorus (P), calcium (Ca), and strontium (Sr), while deeper horizons were enriched in arsenic (As), lithium (Li), and selenium (Se). Intermediate soil horizons showed elevated concentrations of copper (Cu), manganese (Mn), magnesium (Mg), and lead (Pb). Interannual variation was significant for molybdenum (Mo), phosphorus (P), and strontium (Sr), reflecting environmental or management differences between the two seasons.

**Table 1. Soil ion concentration of the experimental site measured in 2021 and 2022.**

| Element | Year | Min | Max | Mean | sd | CV (%) | Location | Depth effect | Year effect |
|---|---|---|---|---|---|---|---|---|---|
| As | 2021 | 0.16 | 0.50 | 0.32 | 0.09 | 27.63 | 0.80 | **1.68E-10** | **4.22E-12** |
|  | 2022 | 0.02 | 0.04 | 0.03 | 0.01 | 16.99 | 0.40 | **1.24E-09** |  |
| Ca | 2021 | 130.18 | 748.77 | 512.58 | 152.83 | 29.82 | **0.01** | **0.001** | **0.001** |
|  | 2022 | 210.36 | 601.95 | 405.35 | 95.39 | 23.53 | 0.79 | **2.06E-09** |  |
| Co | 2021 | 0.02 | 0.22 | 0.09 | 0.04 | 44.53 | 0.32 | **0.04** | 0.11 |
|  | 2022 | 0.03 | 0.13 | 0.07 | 0.02 | 25.72 | **0.002** | 0.65 |  |
| Cr | 2021 | 0.01 | 0.15 | 0.06 | 0.04 | 65.38 | **0.03** | **2.29E-05** | 0.18 |
|  | 2022 | 0.02 | 0.06 | 0.03 | 0.01 | 33.28 | 0.98 | **4.53E-16** |  |
| Cu | 2021 | 0.29 | 0.66 | 0.46 | 0.08 | 17.97 | **0.00002** | 0.19 | **0.01** |
|  | 2022 | 0.31 | 0.44 | 0.37 | 0.03 | 8.83 | 0.38 | **0.02** |  |
| Fe | 2021 | 7.93 | 24.89 | 14.58 | 4.35 | 29.83 | 0.56 | **8.25E-05** | **4.75E-06** |
|  | 2022 | 4.69 | 10.68 | 7.36 | 1.18 | 16.05 | 0.15 | 0.38 |  |
| K | 2021 | 10.06 | 34.06 | 23.81 | 5.33 | 22.38 | **0.00** | 0.25 | **0.01** |
|  | 2022 | 12.16 | 47.88 | 22.85 | 8.77 | 38.37 | 0.82 | **1.06E-08** |  |
| Li | 2021 | 0.00 | 0.01 | 0.01 | 0.00 | 54.11 | 0.96 | **1.42E-06** | **0.0002** |
|  | 2022 | 0.00 | 0.01 | 0.00 | 0.00 | 89.94 | 0.69 | 0.04 |  |
| Mg | 2021 | 39.94 | 176.12 | 111.74 | 29.25 | 26.18 | **0.01** | 0.04 | **1.75E-08** |
|  | 2022 | 44.87 | 112.61 | 71.35 | 13.39 | 18.77 | 0.50 | **5.10E-05** |  |
| Mn | 2021 | 2.67 | 13.58 | 7.38 | 2.50 | 33.94 | 0.78 | **7.39E-05** | **0.01** |
|  | 2022 | 2.76 | 8.72 | 5.83 | 1.56 | 26.76 | 0.57 | **0.002** |  |
| Mo | 2021 | 0.00 | 0.01 | 0.00 | 0.00 | 42.93 | 0.07 | **0.004** | 0.48 |
|  | 2022 | 0.00 | 0.01 | 0.00 | 0.00 | 35.58 | 0.47 | **0.03** |  |
| Na | 2021 | 29.52 | 140.82 | 65.23 | 29.05 | 44.54 | **0.003** | 0.06 | **0.0002** |
|  | 2022 | 15.74 | 74.87 | 39.09 | 13.11 | 33.54 | 0.78 | **0.0001** |  |
| Ni | 2021 | 0.09 | 0.64 | 0.22 | 0.12 | 54.06 | 0.89 | **0.00031** | **0.01** |
|  | 2022 | 0.06 | 0.20 | 0.11 | 0.03 | 29.75 | 0.79 | **4.45E-09** |  |
| P | 2021 | 0.74 | 24.86 | 7.74 | 7.05 | 91.15 | 0.99 | **1.52E-19** | 0.82 |
|  | 2022 | 1.22 | 24.83 | 7.71 | 6.59 | 85.44 | 0.92 | **7.77E-17** |  |
| Pb | 2021 | 0.15 | 0.38 | 0.27 | 0.05 | 18.38 | 0.75 | **0.0002** | **7.68E-07** |
|  | 2022 | 0.11 | 0.26 | 0.18 | 0.04 | 22.12 | 0.10 | **0.0023** |  |
| Rb | 2021 | 0.24 | 0.71 | 0.46 | 0.15 | 33.07 | 0.07 | **4.51E-06** | **0.0001** |
|  | 2022 | 0.12 | 0.58 | 0.31 | 0.12 | 38.56 | 0.08 | **8.72E-05** |  |
| Se | 2021 | 0.97 | 3.07 | 1.94 | 0.57 | 29.58 | 0.98 | **3.75E-11** | **1.20E-12** |
|  | 2022 | 0.01 | 0.02 | 0.01 | 0.00 | 29.81 | 0.89 | **1.73E-13** |  |
| Sr | 2021 | 2.79 | 27.53 | 12.30 | 7.67 | 62.33 | 1.00 | **4.51E-26** | 0.36 |
|  | 2022 | 3.19 | 17.66 | 9.84 | 4.66 | 47.38 | 0.86 | **5.87E-15** |  |
| Zn | 2021 | 0.12 | 2.35 | 0.50 | 0.49 | 97.61 | 0.66 | **1.10E-06** | 0.42 |
|  | 2022 | 0.10 | 1.56 | 0.47 | 0.41 | 87.88 | 0.82 | **1.92E-14** |  |

Mean, standard deviation (SD), coefficient of variation (CV), minimum (Min), and maximum (Max) values are provided. Depth and year effects on measured variables are included, with significant $p$-values from the Kruskal-Wallis test indicated in bold.

Correlations between ion concentrations across soil layers were investigated to better understand patterns of nutrient distribution (S2 Fig in S1 File). In 2021, sodium (Na), rubidium (Rb), lithium (Li), arsenic (As), and selenium (Se) concentrations were positively correlated ($R^2 = 0.4$–0.9) and negatively correlated with phosphorus (P), zinc (Zn), chromium (Cr), and strontium (Sr; $R^2 = -0.8$ to -0.3). In 2022, potassium (K), chromium (Cr), phosphorus (P), and zinc (Zn) exhibited

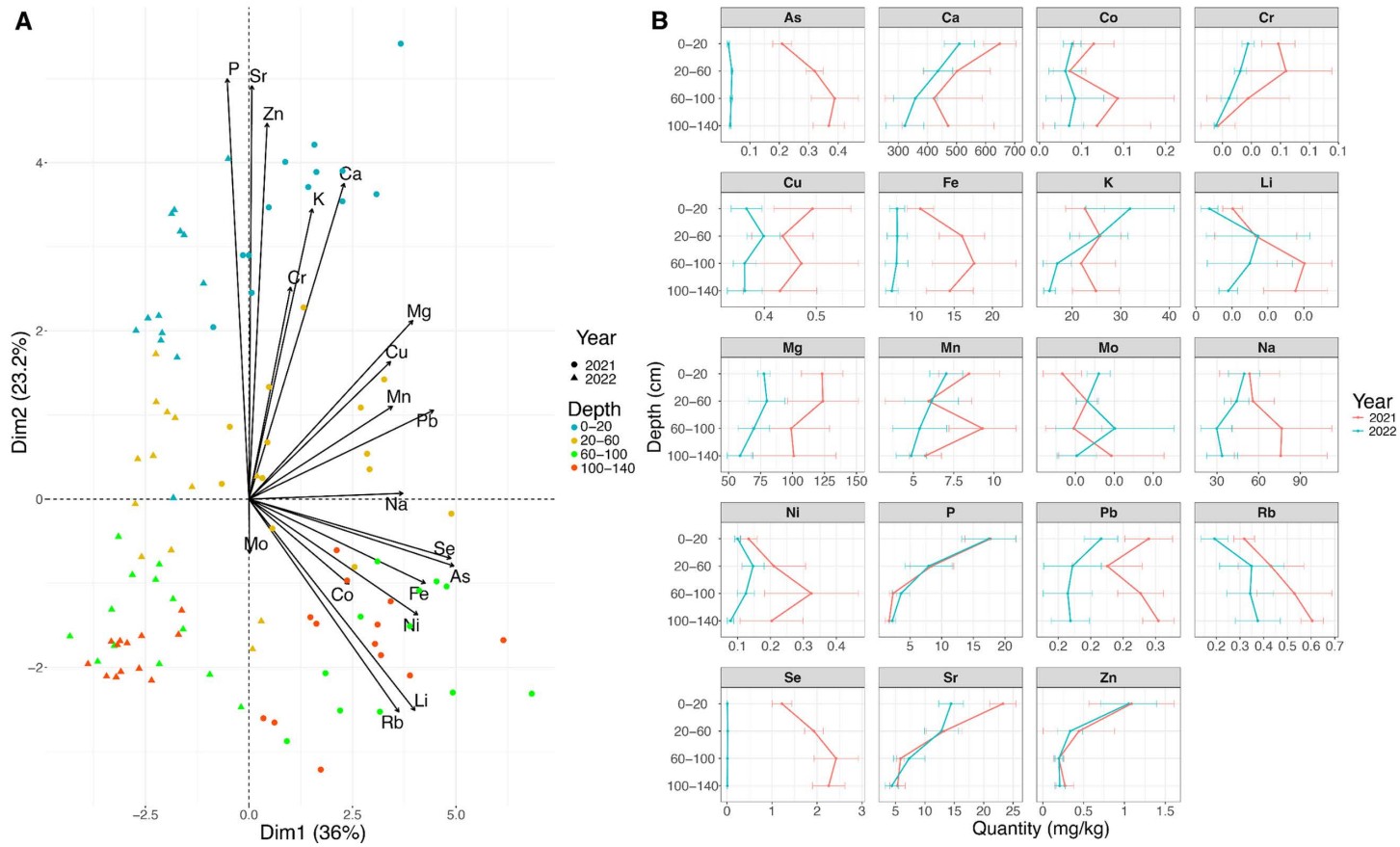

**Fig 1. Soil ion concentration profiles at the field sites.** (A) Principal component analysis (PCA) of soil ion content. The percentage of variance explained by the first two dimensions is indicated on the plot. Colours and shapes indicate different depths and years of sample distribution, respectively. (B) Plot representing soil ion profiles for each ion across years, with the x-axis showing concentration in mg/kg and the y-axis showing depth in cm.

positive correlations (R² = 0.5–0.8) and were negatively correlated with rubidium (Rb), lithium (Li), and arsenic (As; R² = -0.7 to -0.4).

Altogether, the analysis of soil ion content revealed notable differences in nutrient distribution between the 2021 and 2022 field sites. However, certain ions like P or Zn exhibited consistent stratification patterns across years, suggesting stable depth-dependent trends in nutrient accumulation.

### Ion content variability in pearl millet leaves

We characterized the phenotypic variability of ion accumulation in pearl millet leaves for 17 ions (Ca, Cd, Co, Cu, Fe, K, Li, Mg, Mn, Mo, Na, Ni, P, Rb, S, Sr, Zn; Table 2). Across both years, macronutrients (K, Ca, Mg, P, S) dominated the ion profiles, with potassium (K) showing the highest concentration. Among the micronutrients, molybdenum (Mo) and nickel (Ni) had the lowest concentrations. Principal Component Analysis (PCA) revealed a clear separation of ion accumulation patterns by year, indicating a significant influence of the field site on ion content (Fig 2). The first five principal components accounted for 79% of the total variance, with the first axis (Dim.1) alone explaining 49.4%. Dim.1 was strongly driven by Fe, Cu, Ni, Na, and Mg, all of which showed high contribution and projection quality (cos² > 0.5). These ions are associated with oxidative stress response, electron transport, and ionic homeostasis, suggesting that this axis reflects core

**Table 2. Phenotypic variation of leaf element content (in mg/kg) in the PMIGAP panel measured in 2021 and 2022.**

| Element | Year | Min | Max | Mean | sd | H² | CV(%) | Year Effect |
|---------|------|-----|-----|------|-----|-----|-------|-------------|
| Ca | 2021 | 5047.78 | 24114.23 | 10918.03 | 4211.12 | 0.92 | 38.57 | **4.84E-19** |
|    | 2022 | 4229.53 | 12461.66 | 7319.94 | 1662.28 | 0.74 | 22.71 | |
| Cd | 2021 | 0.08 | 1.05 | 0.42 | 0.18 | 0.82 | 42.99 | **0.00462** |
|    | 2022 | 0.22 | 0.9 | 0.46 | 0.14 | 0.6 | 30.35 | |
| Co | 2021 | 0.03 | 0.19 | 0.08 | 0.02 | 0.76 | 31.17 | **1.62E-15** |
|    | 2022 | 0.02 | 0.15 | 0.06 | 0.02 | 0.57 | 37.5 | |
| Cu | 2021 | 4.29 | 13.71 | 8.79 | 1.78 | 0.85 | 20.28 | **4.80E-27** |
|    | 2022 | 3.3 | 10.13 | 6.34 | 1.2 | 0.68 | 18.97 | |
| Fe | 2021 | 53.88 | 176.74 | 98.42 | 25.81 | 0.8 | 26.23 | **4.94E-28** |
|    | 2022 | 35.38 | 131.26 | 64.83 | 16.85 | 0.76 | 25.99 | |
| K | 2021 | 13176.36 | 61540.24 | 31146.88 | 8203.5 | 0.85 | 26.34 | **4.92E-18** |
|   | 2022 | 18419.91 | 64398.36 | 40960.6 | 9337.44 | 0.65 | 22.8 | |
| Li | 2021 | 0.02 | 0.13 | 0.06 | 0.02 | 0.75 | 37.93 | **6.26E-34** |
|    | 2022 | 0 | 0.07 | 0.03 | 0.01 | 0.26 | 39.29 | |
| Mg | 2021 | 3572.68 | 12903.53 | 7511.89 | 1875.25 | 0.76 | 24.96 | 0.107 |
|    | 2022 | 4637.59 | 11957.73 | 7734.71 | 1421.35 | 0.62 | 18.38 | |
| Mn | 2021 | 17.84 | 120.08 | 56.15 | 19.59 | 0.76 | 34.9 | **4.15E-16** |
|    | 2022 | 16.65 | 71.6 | 39.47 | 9.24 | 0.53 | 23.41 | |
| Mo | 2021 | 0 | 3.39 | 1.3 | 0.68 | 0.87 | 52.74 | **0.000392** |
|    | 2022 | 0.28 | 2.66 | 0.99 | 0.38 | 0.8 | 38.38 | |
| Na | 2021 | 248.44 | 3161.79 | 1212.4 | 498.06 | 0.85 | 41.08 | **6.57E-06** |
|    | 2022 | 641.52 | 4147.26 | 1435.61 | 457.84 | 0.78 | 31.89 | |
| Ni | 2021 | 0.66 | 2.81 | 1.74 | 0.41 | 0.21 | 23.74 | **5.39E-39** |
|    | 2022 | 0.26 | 2.05 | 0.8 | 0.27 | 0.29 | 33.13 | |
| P | 2021 | 2351.39 | 7175.52 | 4388.97 | 796.04 | 0.74 | 18.14 | **8.94E-47** |
|   | 2022 | 1273.94 | 3227.2 | 2124.36 | 396.11 | 0.59 | 18.65 | |
| Rb | 2021 | 10.13 | 63.13 | 39.35 | 8.12 | 0.79 | 20.64 | **4.08E-11** |
|    | 2022 | 24.64 | 74.51 | 46.4 | 8.69 | 0.6 | 18.73 | |
| S | 2021 | 1729.89 | 4454.19 | 2871.25 | 403.24 | 0.81 | 14.04 | **8.36E-40** |
|   | 2022 | 1300.36 | 2933.75 | 2058.78 | 292.19 | 0.58 | 14.19 | |
| Sr | 2021 | 135.25 | 646.39 | 296.39 | 105.87 | 0.9 | 35.72 | **0.0274** |
|    | 2022 | 129.71 | 478.55 | 259.6 | 59.32 | 0.63 | 22.85 | |
| Zn | 2021 | 28.4 | 163.26 | 65.12 | 16.94 | 0.65 | 26.02 | **6.61E-36** |
|    | 2022 | 15.04 | 73.1 | 38 | 9.14 | 0.62 | 24.06 | |

Mean, standard deviation (SD), coefficient of variation (CV), minimum (Min), maximum (Max), and heritability (H²) values are provided. Year effects on measured variables are included, with significant *p*-values from the Kruskal-Wallis test indicated in bold.

physiological processes influenced by both genotype and environmental conditions. Dim.2 (10.7%) was mainly structured by Mg, K, and Rb, with high contributions and reliable cos² values. These elements are central to osmotic regulation and nutrient transport, highlighting a strong environmental influence on ion uptake, possibly related to water availability or soil composition differences between years. Inter-annual differences were statistically significant for all ions except magnesium (Mg; Table 2, S3 Fig in S1 File).

A broad range of variability in leaf ion content was observed among the genotypes. Most ions, except for sulphur (S) and phosphorus (P), exhibited a coefficient of variation (CV) exceeding 19% (Table 2). In 2021, the highest phenotypic

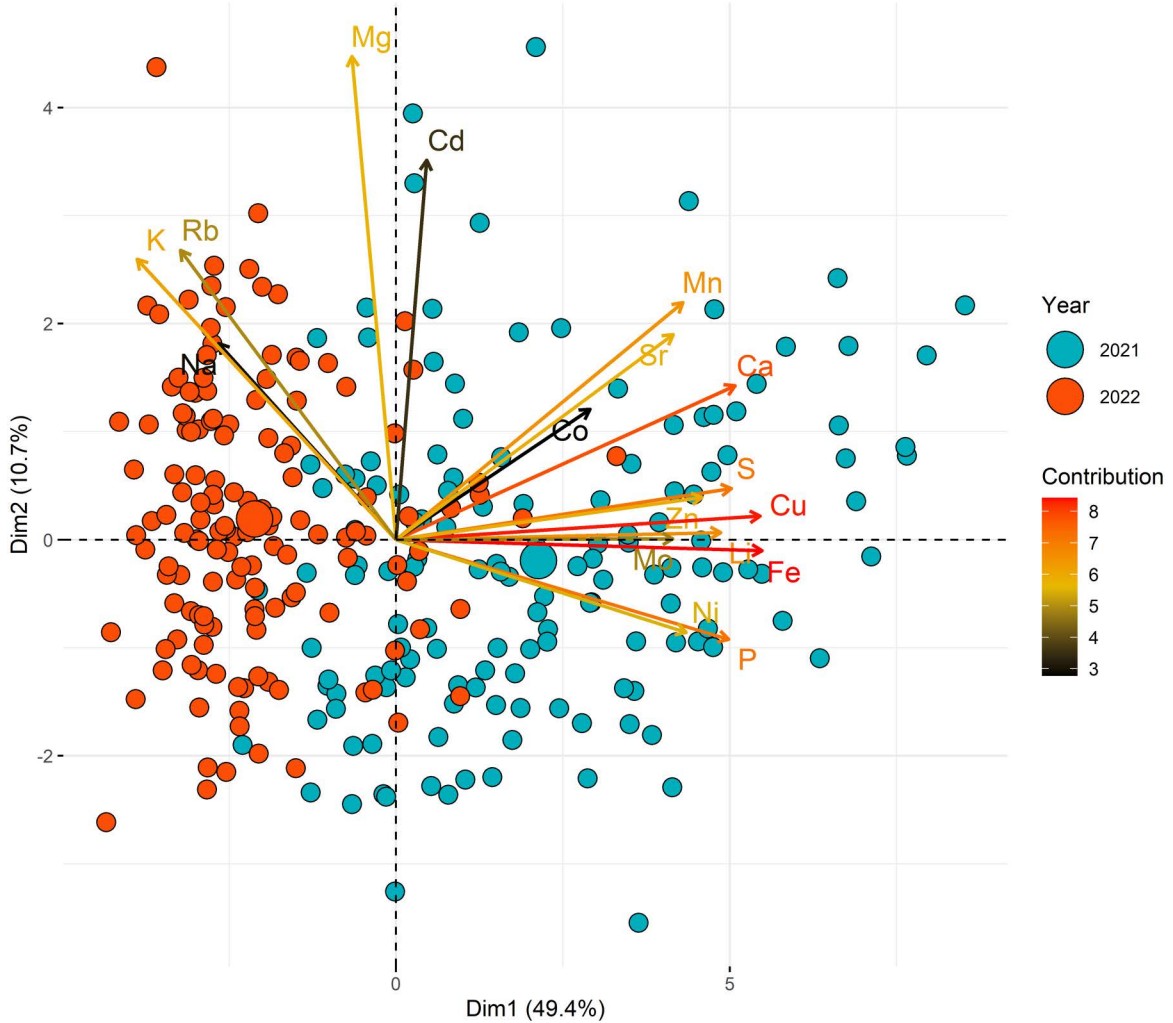

**Fig 2. Principal component analysis (PCA) of BLUEs measured both years for leaf ion content in the PMIGAP panel.** Principal Component Analysis (PCA) of leaf ion concentrations in 2021 (blue) and 2022 (red). Dimension 1 (49.4%) and Dimension 2 (10.7%) capture most of the variance. Arrows indicate ion contributions, with length showing strength and colour (black to red) representing cos² values. Points represent genotypes, highlighting year-based differences.

variation was observed for molybdenum (Mo) with a CV of 52.7%, followed by cadmium (Cd; CV = 43.0%) and sodium (Na; CV = 41.1%). In 2022, lithium (Li) showed the highest variability (CV = 39.3%), followed by molybdenum (Mo; CV = 38.4%) and cobalt (Co; CV = 37.5%).

Broad-sense heritability estimates for ion concentrations reveal a substantial genetic contribution to phenotypic variation for most ions, with values ranging from 0.65 to 0.92 in 2021 and from 0.53 to 0.80 in 2022 (Table 2). However, for certain ions like nickel (Ni) and lithium (Li), heritability was notably lower (0.21–0.29), indicating that environmental factors had a greater influence on their phenotypic variability.

To explore ion accumulation patterns, Pearson's correlation analyses were conducted (S4 Fig in S1 File). The results revealed moderate to very strong correlations, consistent across both 2021 and 2022. Significant positive correlations were observed among calcium (Ca), strontium (Sr), molybdenum (Mo), phosphorus (P), and copper (Cu) (R² = 0.20–0.98). In contrast, these ions exhibited negative correlations with potassium (K) and rubidium (Rb) (R² = -0.49 to -0.18). The

stability of these associations across two growing seasons suggests that these relationships are robust under varying environmental conditions. These correlations between ion concentration could be explained by different mechanisms such as similar accumulation in the soil profile or common transport mechanisms for example.

This analysis highlights the complex interplay between genetic and environmental factors in determining ion content in leaves, while also uncovering stable inter-ion relationships that may inform future studies on nutrient transport and accumulation.

### Relationship between leaf ion content and agro-morphological and root traits

We next analysed the correlations between leaf ion content and agro-morphological and root traits measured in the same field trials [30]. Only correlations consistently observed across the two years of the study (2021 and 2022) were considered.

We first investigated the relationship between phenology and leaf ion content. Across both years, significant positive correlations were observed between flowering time and the concentrations of sodium (Na), magnesium (Mg), potassium (K), and rubidium (Rb). In contrast, manganese (Mn), iron (Fe), copper (Cu), and molybdenum (Mo) displayed significant negative correlations with flowering time ($p$-value < 0.05; Fig 3).

Total grain weight at harvest was not correlated with phenological parameters. However, it showed a positive correlation ($R^2$ = 0,23–025) with manganese (Mn) content in the leaves at 49 DAS in 2021 and 42 DAS in 2022 (Fig 3),

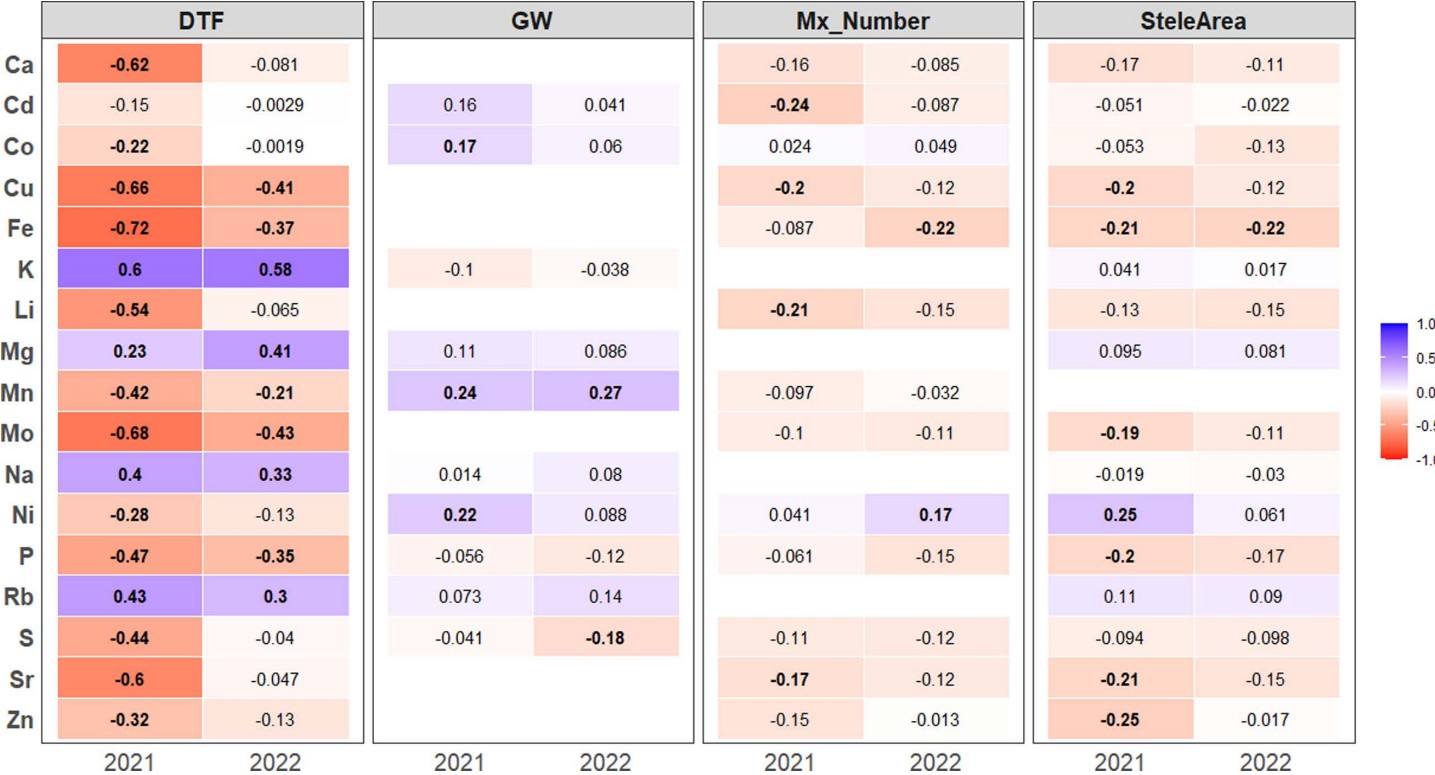

**Fig 3. Correlation between ion content, root traits, and agro-morphological traits in 2021 and 2022.** Heatmap illustrating Pearson's correlation coefficients between ion content, root traits (Mx_Nbr: metaxylem vessels number, SteleArea: stele area), and agro-morphological traits (DTF: days to flowering, GW: total grain weight). The color gradients represent the magnitude of Pearson's correlation coefficients, while significant correlations ($p$ > 0.05) are marked in bold. Only correlations consistently observed across the two years of the study (2021 and 2022) are represented.

suggesting that optimal Mn content in leaves is critical for maximizing plant growth, biomass production, and ultimately, grain yield in cereals.

In terms of root anatomical traits, consistent negative correlations were identified between stele area and iron (Fe; Fig 3). We also found significant correlation between metaxylem vessel number and Fe concentrations in 2022 (Fig 3). Interestingly, metaxylem vessel number and stele area were positively correlated (S5 Fig in S1 File). These findings suggest that increasing the number of xylem vessels may directly or indirectly reduce the acquisition of iron.

Our results highlight the complex interplay between leaf ion content, phenological traits, and root anatomy in cereals in field conditions. The correlations between root traits and ion content reveal potential trade-offs in nutrient acquisition strategies.

### Identification of genomic regions associated with leaf ion content in pearl millet

To identify genomic regions controlling leaf ion accumulation, we conducted a genome-wide association study (GWAS) using 254,765 high-quality single nucleotide polymorphisms (SNPs) as genotypic data, and Best Linear Unbiased Estimates (BLUEs) as phenotypic values for 16 ions (excluding Ni due to low heritability). The BLUEs exhibited a normal distribution (S6 Fig in S1 File). A total of 359 marker-trait associations were identified using a 5% adjusted False Discovery Rate (FDR) threshold across four GWAS methods (LFMM, EMMA, BLINK, and CMLM; S3 Table in S1 File). Quantile-Quantile (QQ) plots indicated that the GWAS models fitted well to the data, with observed $p$-values distributed uniformly and showing inflation at higher values, indicative of true genetic signals (S7 Fig in S1 File).

To minimize spurious associations and environmental noise, we prioritized associations detected by at least two GWAS models in one year and confirmed using Fisher's combining method across both years. This stringent approach identified 78 common marker-trait associations (S4 Table in S1 File). The majority of markers were associated with cobalt (Co; 67 SNPs) across all chromosomes except chromosome 6. Additional associations were detected for potassium (K; five SNPs) on chromosomes 2, 3, and 7; magnesium (Mg; three SNPs) on chromosome 3 and the unknown chromosome (ChrUn); cadmium (Cd; one SNP, S7 Fig in S1 File) on chromosome 7; iron (Fe; one SNP) on ChrUn; and sulfur (S; one SNP) on chromosome 1 (S4 Table in S1 File).

Pairwise linkage disequilibrium (LD) analysis was performed to delineate quantitative trait loci (QTLs) based on significant SNPs. A total of 43 QTLs were identified (S5 Table in S1 File). No colocalization was observed between QTLs for different ions, suggesting that the genetic mechanisms underlying ion accumulation are independent (S5 Table in S1 File). Hence, we identified genomic regions associated with the accumulation of various ions in pearl millet leaves. The lack of colocalization between QTLs for different ions suggests that these QTLs correspond to independent genetic pathways despite the fact that the concentration of some of these ions were correlated.

### Impact of flowering time as a confounding factor in GWAS of leaf ion content

Leaf concentration of ions might be partially associated with the development stage. We notably observed in some cases a correlation between flowering time and ion content. Phenology may act as a confounding factor in association genetics analyses, potentially masking true genetic associations. To correct for the potential confounding effect of phenology on ion concentration, we replicate the GWAS analysis on the residual of the linear regression between individual ion concentrations and flowering time. The GWAS analysis based on the residuals identified 83 significant SNPs meeting the selection criteria (associations detected by at least two GWAS models in one year and confirmed using Fisher's combining method across both years; S6 Table in S1 File). Of these, 74 SNPs were associated with cobalt (Co) across all chromosomes. Additional associations were observed for magnesium (Mg; two SNPs) on chromosome 3 and the unknown chromosome (ChrUn), phosphorus (P; one SNP) on ChrUn, rubidium (Rb; four SNPs) on chromosome 2 and ChrUn, and strontium (Sr; two SNPs) on chromosome 6. Interestingly, 61 of these SNPs were consistent with those identified using uncorrected

BLUEs (59 SNPs for Co and two SNPs for Mg), while 22 SNPs were unique to the residual-based analysis (15 SNPs for Co, one SNP for P, four SNPs for Rb and two SNPs for Sr). Additionally, we identified 31 SNPs associated with flowering time, three of which co-localized with SNPs associated with potassium (K; S7 Table in S1 File).

A total of 40 QTLs were identified from significant SNPs associated with the residues (S5 Table in S1 File). Among these, the majority are associated with cobalt, with 33 QTLs, of which 23 had already been identified in the previous GWAS, while 10 are new and specific to residues. Additionally, one QTL was detected for phosphorus (P), while strontium (Sr) and rubidium (Rb) each have two QTLs specific to residues. Finally, two QTLs were identified for magnesium (Mg).

Hence, by adjusting for the effects of phenology, we uncovered additional genetic associations, emphasizing the value of accounting for phenotypic relationships in genetic studies.

### Identification of candidate genes for leaf ion content in pearl millet

To identify candidate genes associated with leaf ion content in pearl millet, we analysed genes located within key QTL regions. This analysis integrated published gene expression data from pearl millet roots [62] and leaves [69], newly generated RNA-seq data on different root types (primary root, crown roots and long lateral roots), and sequence homology information (S8 Table in S1 File).

Two QTLs for leaf magnesium (Mg) content were identified in both GWAS analysis using BLUEs and residuals, suggesting independence from plant phenology. QTL Mg_Cont3–4 on Chromosome 3 spans a 50-kb region centred around the most significant SNP at position 177312638 bp (Fig 4). Two genotypes of the SNP were present in our population: C/C (n = 127) and C/T (n = 15). The C/T genotype showed a 25.31% higher phenotypic value compared to the C/C genotype, which is the reference allele. This difference was statistically significant (Fig 4). Although no annotated genes are present within this region, RNA-seq data revealed transcriptional activity, suggesting the presence of an unannotated gene (Fig 4). Sequence homology analysis suggests these transcribed regions encode a putative pyruvate kinase; an enzyme central to glycolysis for which magnesium serves as a critical cofactor. This finding might suggest a potential feedback mechanism linking Mg nutrition and pyruvate kinase activity. The second QTL mapped to a region unassigned to any chromosome in the current genome assembly. Three genotypes of the SNP were present in our population: C/C (n = 99), G/G (n = 35) and C/G (n = 8). The C/C genotype showed a 15.35% lower phenotypic value compared to the G/G genotype, which is the reference allele. This difference was statistically significant. No annotated genes were found in this region, and further studies are required to identify the corresponding functional elements contributing to Mg accumulation.

We next examined the four QTLs associated with leaf potassium (K) content, two of which co-localized with flowering time QTLs (QTL K_Cont2–1 on Chromosome 2 and QTL K_Cont7–5 on Chromosome 7). The most significant SNP in QTL K_Cont2–1 resides within the predicted gene *Pgl_GLEAN_10002101*, which is highly expressed in roots (Fig 5). Three genotypes of the SNP were present in our population: C/C (n = 127), T/T (n = 5) and C/T (n = 7). The C/C genotype showed a 20.97% higher phenotypic value compared to the T/T genotype. Additionally, a significant difference was observed between C/C and C/T genotypes, with C/C showing a 36.98% higher phenotypic value than C/T. These results show that the C allele has a significant effect on the phenotype, increasing its value significantly in the homozygous state compared with the T reference allele and the heterozygous state. This gene encodes a protein with homology to the *Arabidopsis thaliana* Expp1 protein (AT3G44150), a plasma membrane-localized protein of unknown function but conserved across plant species, including maize and rice. QTL K_Cont7–5 on Chromosome 7 contains only one predicted gene, *Pgl_GLEAN_10014595*, starting 20,857 bp downstream of the most significant SNP located at position 136644636. This gene encodes a putative E3 ubiquitin-protein ligase. Moreover, RNA-seq data showed high transcriptional activity 3,792 bp upstream of the most significant SNP. This region shows homology to a pearl millet expressed sequence tag (EST) and an Arabidopsis gene encoding a serine palmitoyltransferase-like subunit (*AT1G06130*), an enzyme involved in sphingolipid biosynthesis, which may influence potassium channel activity or transporters. The remaining two QTLs for K content were independent of flowering time. QTL K_Cont7–4 on Chromosome 7 has no annotated gene. Similarly, no predicted proteins

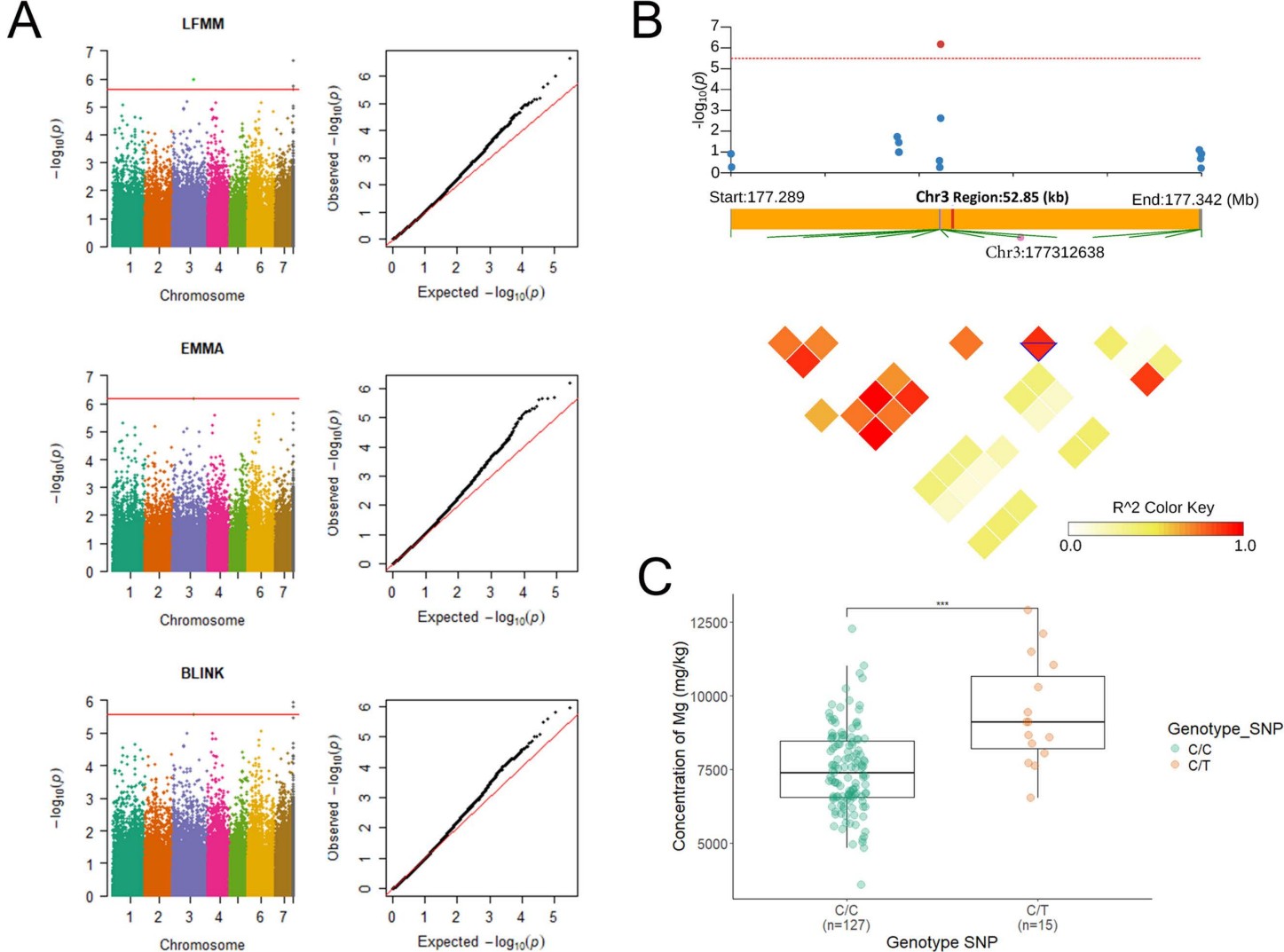

**Fig 4. GWAS result for magnesium.** (A) Manhattan plots (left) and QQ plots (right) presenting the results of the genetic association analysis for magnesium concentration using the Fisher combining method with LFMM, EMMA, and BLINK. The red line indicates the significance threshold for the respective methods. (B) Local Manhattan plot for QTL Mg_Cont3-4 on chromosome 3 and linkage disequilibrium (LD) plot for the QTL region spanning 52.85 kb, with color shading representing R² values. The transcribed region is highlighted in red. (C) Phenotypic value comparison across genotypes for the significant SNP. Boxplots illustrate the medians and distributions of values for each genotype, with significant differences indicated by asterisks. ***: $p$-value < 0.001.

are present within QTL K_Cont3–6 on Chromosome 3, but a region transcribed in roots, located 3.7 kb downstream of the most significant SNP, shows homology to a pearl millet expressed sequence tag (EST; GenBank CD725629.1) and a predicted long non-coding RNA (lncRNA) from *Setaria viridis*.

Cadmium is a toxic heavy metal that can enter the food chain through forage crops consumed by livestock. Interestingly, we found one QTL for leaf cadmium content on chromosome 7 (QTL Cd_Cont7–1) located around the most significant SNP located at position 34865024. We found two annotated genes in this region. The first one, *Pgl_GLEAN_10030409*, is located 8,376 bp downstream of the most significant SNP and the corresponding predicted protein has homologies with glycosyl group transferase. The second gene, *Pgl_GLEAN_10030410*, is located 14,982 bp

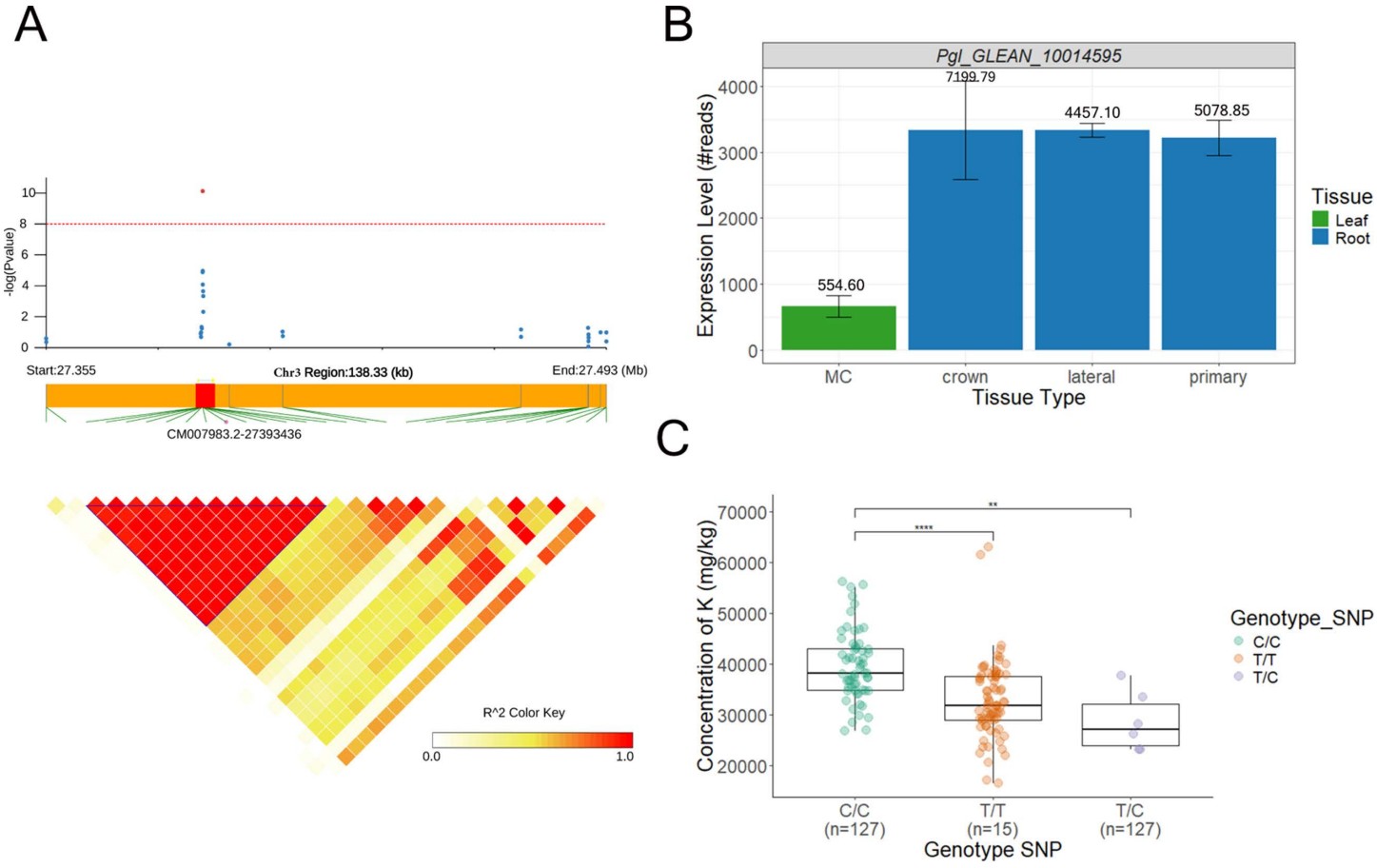

**Fig 5. Genetic association analysis of QTL K_Cont2-1 associated with potassium content.** (A) Local Manhattan plot in QTL K_Cont2–1 region, where each point represents a SNP, and the Y-axis indicates −log10(p) for significance. The red line marks the threshold for statistical significance. The lower panel shows a linkage disequilibrium (LD) plot in a region of 138.33 (Kb), with color shading representing R² values. The LD block highlighted in blue represents 0.6 Kb. *Pgl_GLEAN_10002101* is highlighted in red. (B) *Pgl_GLEAN_10002101* relative expression levels in different tissues from RNAseq data. (C) Comparison of phenotypic values across genotypes for the significant SNP. Boxplots display medians and the distribution of values for genotypes of each allelic group, with significant differences denoted by asterisks.

downstream of the most significant SNP and shows homology with a gene encoding an *A. thaliana* protein localised in the Golgi apparatus (*AT5G66030*) potentially involved in vesicular transport.

## Discussion

In this study, we investigated the factors influencing leaf ion content in pearl millet under field conditions, aiming to identify determinants that could be targeted to improve nutrient use efficiency. Besides, pearl millet is a critical source of forage in arid and semi-arid regions of sub-Saharan Africa and India, and ion content directly impacts forage quality by increasing micronutrient concentrations (e.g., Fe, Zn) and reducing toxic compounds (e.g., Cd). Our findings reveal the diversity in ion acquisition, transport, and storage strategies in pearl millet leaves, highlighting complex interactions between soil properties, root traits, and leaf ion concentrations.

Soil ion profiles exhibited significant variations with depth and between years, driven by pedological and climatic processes that influence nutrient availability. Ion absorption by roots is influenced by their availability in the soil, which depends on factors such as aeration, pH and solubility [70]. Roots use both active and passive mechanisms for nutrient

absorption [9]. Once absorbed, ions are transported to the leaves via the xylem [9,70,71]. Leaf ion concentrations varied significantly between years and genotypes, reflecting the combined effects of environmental and genetic factors on nutrient acquisition and transport efficiency. The substantial variation among genotypes, coupled with high heritability, underscores the significant role of genetic variation in nutrient acquisition and efficiency. The strong vertical stratification of ion concentrations in the soil, where surface layers are enriched in elements like calcium (Ca), potassium (K), and phosphorus (P), aligns with the higher concentrations of these elements in pearl millet leaves. However, the relationship between soil ions and leaf concentrations is not solely governed by their abundance in the soil. Root uptake efficiency, ion transport mechanisms, and root-soil interactions are key factors that modulate the availability of these ions to the plant [70,71]. The observed positive correlations between ions such as P, Zn, Ca, and Sr in both soil and leaves support the idea of shared transport pathways or similar absorption processes. Conversely, the negative correlations between surface-bound ions (P) and those associated with deeper soil layers (As) may reflect distinct uptake mechanisms or competition for resources. We therefore conducted genome-wide association studies (GWAS) to identify genomic regions controlling leaf ion content in pearl millet. Our analysis revealed several potential quantitative trait loci (QTLs) and, in some cases, candidate genes. For example, a single nucleotide polymorphism (SNP) associated with magnesium (Mg) content suggested a potential functional link between Mg and pyruvate kinase, a critical enzyme in plant energy metabolism. This QTL remained robust even after accounting for phenology, indicating a strong association. Pyruvate kinase catalyses the final step of glycolysis, converting phosphoenolpyruvate (PEP) into pyruvate, with magnesium acting as an essential cofactor [72]. While the role of magnesium in pyruvate kinase activity is well established, our findings suggest a potential feedback mechanism between this enzyme and Mg acquisition, which warrants further investigation.

Another notable result was the identification of a SNP associated with potassium (K) accumulation near a gene encoding a serine palmitoyltransferase-like subunit, implicating sphingolipids in plant mineral nutrition. Sphingolipids, as critical components of plant membranes, influence their fluidity and organization, potentially affecting the activity or localization of K+ channels [73]. Mutations in sphingolipid biosynthesis genes have been shown to alter nutrient profiles, including K, Mg, and Fe, in Arabidopsis [11]. Additionally, sphingolipids may act as signalling molecules, regulating stress responses, growth, and development, which could indirectly influence K uptake, transport, or compartmentalization.

We also identified a SNP associated with potassium near a region homologous to a potential long non-coding RNA (lncRNA). LncRNAs are known to regulate gene expression, including genes involved in potassium absorption and transport, such as those encoding KT/KUP/HAK family transporters [74,75]. These findings emphasize the importance of exploring non-coding genomic regions to uncover regulatory mechanisms underlying nutrient use efficiency.

Finally, we identified one QTL for cadmium accumulation in the leaves. Limiting cadmium concentration in plants is crucial due to its potential toxicity to both plants and animals. Cadmium, a heavy metal, can accumulate in plant tissues and disrupt essential physiological processes, leading to reduced growth, chlorosis, and even plant death [76]. Consumption of cadmium-contaminated forage crops by animals could lead to cadmium accumulation in the food chain, eventually resulting in health issues [77]. This QTL could be used to reduce cadmium levels in pearl millet biomass, an important source of forage in the drylands, thereby ensuring food safety and protecting both environmental and human health.

Many SNPs were associated with cobalt (Co) content in the leaves in our GWAS study, which is surprising given that cobalt is present in low concentrations compared to other micronutrients. Cobalt is not a major element for plants [78] and has only minor importance in forage as a component of vitamin B12, which is necessary for the proper functioning of the nervous system and the production of red blood cells in animals [79]. On the other hand, excessive levels can be toxic. We have no clear explanation for this high number of marker trait associations (MTAs).

Leaf ion content correlations with agromorphological traits also provided important insights. A positive correlation was observed between manganese (Mn) concentration in leaves during the vegetative phase and grain yield. Mn is a critical cofactor for enzymes involved in photosynthesis, respiration, and nitrogen metabolism, processes essential for plant growth and biomass production [80,81]. Additionally, Mn plays a role in antioxidative defence, helping mitigate oxidative

stress caused by environmental factors such as drought and heat. These results suggest that Mn could be a limiting factor for pearl millet growth in field conditions. However, we did not identify any QTLs controlling Mn leaf content in our GWAS analysis or find correlations with root traits that could be leveraged to improve Mn nutrition.

Root anatomical traits were also linked to leaf ion content. Xylem size and number is important for water acquisition in pearl millet [30]. However, negative correlations between metaxylem vessel number, stele area and Fe concentration were observed suggesting a trade-off in resource allocation. For example, increased stele size may indicate a shift in resource allocation toward root growth to capture water, potentially at the expense of Fe uptake mechanisms. Alternatively plants with more xylem vessels could take up more water relative to Fe uptake and develop more biomass, thus leading to a dilution of the concentration of this ion in the leaves. Environmental factors such as drought and salinity could further influence these relationships, complicating Fe homeostasis.

Finally, correlations between ion concentrations in leaves revealed additional interactions. The positive correlation between calcium (Ca) and strontium (Sr) likely reflects their shared transport mechanisms, while the correlation between Ca and phosphorus (P) points to their joint involvement in membrane formation and tissue structure [82]. In contrast, negative correlations, such as that between potassium (K) and rubidium (Rb), may indicate competition for transport sites or antagonistic interactions [9]. Despite these correlations, we did not find QTLs controlling more than one ion, suggesting that we detected genomic regions specifically regulating the accumulation of a single ion. This might be explained by various factors, such as the limited number of QTLs we detected or the fact that correlations in leaf concentrations might not be due to common transport mechanisms (e.g., co-localization in the soil profile). Understanding these interactions is critical for optimizing nutrient management practices.

## Conclusion

This study highlights the genetic, physiological, and environmental factors influencing nutrient use efficiency in pearl millet, providing valuable insights for improving its productivity. The identification of QTLs and candidate genes offers promising targets for breeding programs aimed at optimizing ion acquisition, transport, and storage. Additionally, the observed correlations between leaf ion content, agromorphological traits, and root anatomy underline the importance of integrating multiple traits to develop resilient varieties. By advancing our understanding of nutrient use efficiency, this work could contribute to the development of pearl millet varieties better suited to the challenges of climate change, enhancing food and forage security in vulnerable regions. Further studies are needed to validate the identified QTLs and unravel the underlying molecular mechanisms.

## Supporting information

**S1 File.** **S1 Fig**. Field trial experimental design. The design follows a completely randomized block layout with four replicates. Each replicate consists of 10 sub-blocks, each containing 16 plots. Each plot (detail shown in the top right) is planted with three rows of 10 plants of the same genotype, with a spacing of 0.9 m between rows and 0.3 m between plants. The red dots indicate locations where soil samples were collected at 4 different depths (0–140 cm). Leaf samples, soil analyses, and root traits were also studied (illustrations shown at the bottom right). The blue icons represent irrigation pumps. **S2 Fig.** Correlation plot for soil ion content in 2021 (A) and 2022 (B) field sites. Heatmap representing Pearson's correlation coefficients between soil ion concentrations measured at four different depths (0–140 cm). Color gradients indicate the Pearson's correlation coefficient. Non-significant correlations at a $p$-value threshold of 0.05 are indicated with a cross. **S3 Fig.** Boxplot representing variation in ion content in 2021 and 2022 in the PMIGAP panel. Ion content is represented as mg/kg. $p$-values from the Wilcoxon test are represented. **S4 Fig.** Correlation plot for leaf ion content of the PMIGAP panel measured during the experimental field study. Heatmap representing Pearson's correlation coefficients between BLUEs of all accessions of the panel observed in 2021 (A) and 2022 (B). Color gradients indicate the Pearson's correlation coefficient. Non-significant

correlations at a *p*-value threshold of 0.05 are indicated with a cross. **S5 Fig.** Correlation plot for ion content, root (A) and agro-morphological (B) traits in 2021 and 2022. Heatmap representing Pearson's correlation coefficients between ion content, root and agro-morphological traits. Color gradients indicate the Pearson's correlation coefficient. Significant correlations at a *p*-value threshold of 0.05 are indicated in bold. Roots traits are number of metaxylem vessels (MX_Number), mean area of metaxylem vessels (Meansize_MX), sclerenchyma pixel sum (SCL_Area), total area of the root section (RootArea, µm²), total area of metaxylem vessels (Totalsize_MX, µm²), total area of the stele (SteleArea, µm²), Ratio between stele area and root area (SR_Ratio), ratio between SCL and root area (SCL_Ratio). Agromorphological traits are: number of days after sowing when 50% of plants in the plot show flowering (DTF), 1000 grains weight (PMG, g), total grain weight at harvest (GW, g), number of tillers on three plants measured at maturity (Tiller_number), shoot dry biomass of plants phenotyped for root traits (SDW, g) at 49 DAS in 2021 or 42 DAS in 2022, shoot dry biomass of three plants harvested at maturity (SDW_Maturity, g), plant height from soil to flag leaf at maturity (HSDF, cm). **S6 Fig.** Histogram representing variation in ion content in 2021 (A) and 2022 (B) in the PMIGAP panel. Ion content is represented as mg/kg. Data used for the GWAS analysis follow a normal distribution for all the variables studied. **S7 Fig.** Manhattan plot and Quantile-Quantile (QQ) plots of Cadmium. GWAS result using the Fisher combining method with LFMM, EMMA, and BLINK. The red line indicates the significance threshold for the respective methods. QQ Plot indicated that the GWAS models fitted well to the data, with observed p-values distributed uniformly and showing inflation at higher values. **S1 Table.** Passport data of the different pearl millet lines used in the study. **S2 Table.** List of all SNPs identified by GWAS. Information includes the name of the ion (Ion), SNP identifier (SNP), chromosome number (Chrom), SNP position on the chromosome (POS), most significant *p*-value (pvalue_max), method with the most significant *p*-value (significant_method), names of all methods that detected the SNP (methods), and whether the SNP was detected in 2021, 2022, using the combined Fisher method, or all (Year). **S3 Table.** List of marker trait association (MTA) retained. Information includes the name of the ion (Ion), SNP identifier (SNP), chromosome number (Chrom), SNP position on the chromosome (POS), most significant *p*-value (pvalue_max), method with the most significant *p*-value (significant_method), names of all methods that detected the SNP (methods), and whether the SNP was detected in 2021, 2022, using the combined Fisher method, or all (Year). **S4 Table.** List of QTL obtained based on linkage disequilibrium. Information includes the name of the quantitative trait loci (QTL_name), chromosome number (chrom), number of significant MTA in the QTL (nbr Sig_snp), region of the QTL (start and end of the QTL in 50 kb around the most extreme **or** significant MTA; QTL_Pos), variable name used for GWAS of variables without (content: cont) or with (residual: res) flowering time influence, whether detected QTL is identified by GWAS of the ion or residuals, or both (Type), the ion name (Ion). **S5 Table.** List of marker trait association (MTA) retained based on criteria selection and identified by GWAS on residuals of the linear regression between ion content and flowering time. Information includes the name of the ion (Ion), SNP identifier (SNP), chromosome number (Chrom), SNP position on the chromosome (POS), most significant *p*-value (pvalue_max), method with the most significant *p*-value (significant_method), names of all methods that detected the SNP (methods), and whether the SNP was detected in GWAS of the ion or residuals, or both (Type). **S6 Table.** List of marker trait association (MTA) retained based on criteria selection and identified by GWAS on flowering time. Information includes the name of the trait (Day_flowering), SNP identifier (SNP), chromosome number (Chrom), SNP position on the chromosome (POS), most significant *p*-value (pvalue_max), method with the most significant *p*-value (significant_method), names of all methods that detected the SNP (methods). **S7 Table.** Gene Expression level in leaves and roots. Information includes gene name (Gene) and his level expression in leaves (Leaf), in roots (crown, lateral and primary).
(ZIP)

## Acknowledgments

Technical support for the soils and plant analysis was provided by Lolita Wilson, Kenneth Davis and Ibrahim Haji at the Elemental Analysis Facility, School of Biosciences, University of Nottingham.

## Author contributions

**Conceptualization:** Yves Vigouroux, Alexandre Grondin, Laurent Laplaze.

**Data curation:** Elizabeth Bailey, Alexandre Grondin, Laurent Laplaze.

**Formal analysis:** Princia Nakombo-Gbassault, Sebastian Arenas, Pablo Affortit, Awa Faye, Bassirou Sine, Pascal Gantet, Ephrem Kosh Komba, Ndjido Kane, Malcolm Bennett, Darren Wells, Philippe Cubry, Elizabeth Bailey, Yves Vigouroux, Alexandre Grondin, Laurent Laplaze.

**Funding acquisition:** Ndjido Kane, Malcolm Bennett, Alexandre Grondin, Laurent Laplaze.

**Investigation:** Princia Nakombo-Gbassault, Sebastian Arenas, Pablo Affortit, Awa Faye, Paulina Flis, Daniel Moukouanga, Darren Wells, Elizabeth Bailey, Yves Vigouroux, Alexandre Grondin, Laurent Laplaze.

**Project administration:** Alexandre Grondin, Laurent Laplaze.

**Resources:** Philippe Cubry.

**Supervision:** Bassirou Sine, Ephrem Kosh Komba, Ndjido Kane, Darren Wells, Philippe Cubry, Yves Vigouroux, Alexandre Grondin, Laurent Laplaze.

**Validation:** Yves Vigouroux, Alexandre Grondin, Laurent Laplaze.

**Writing – original draft:** Princia Nakombo-Gbassault, Yves Vigouroux, Alexandre Grondin, Laurent Laplaze.

**Writing – review & editing:** Princia Nakombo-Gbassault, Sebastian Arenas, Pablo Affortit, Darren Wells, Elizabeth Bailey, Yves Vigouroux, Alexandre Grondin, Laurent Laplaze.

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
