## [Decision Letter · Decision Letter 0]

19 Mar 2025

PONE-D-25-04942Genetic control of the leaf ionome in pearl millet and correlation with root and agromorphological traitsPLOS ONE

Dear Dr. Laplaze,

Thank you for submitting your manuscript to PLOS ONE. After careful consideration, we feel that it has merit but does not fully meet PLOS ONE’s publication criteria as it currently stands. Therefore, we invite you to submit a revised version of the manuscript that addresses the points raised during the review process.

Please consider the comments of both reviewers while revising the manuscript. Specifically, the reviewers asked for materials and methods to be concise without compromising the quality and discussion should be throughly revised with incorporation of recent references. Please provide high resiolution figures as indicated by the reviewers.

We look forward to receiving your revised manuscript.

Kind regards,

Prasanta K. Subudhi, Ph.D.

Academic Editor

PLOS ONE

Journal Requirements:

3. Thank you for stating the following financial disclosure: [P.N.G. was supported by a joint PhD grant from the Institut de Recherche pour le Développement (IRD, France) and the French Embassy in the Central African Republic. We acknowledge support from the Royal Society (Anatomics grant ICA-R1-180356 to MB and 785 NK), the Agropolis Fondation (ValoRoot grant n°2202-002 to LL) as part of the "Investissement d'avenir" (ANR-l0-LABX-0001-0l), under the frame of I-SITE MUSE (ANR-16-IDEX-0006), and the European Plant Phenotyping Network (EPPN2020 project Ionomil n°386).]. 

Additional Editor Comments:

Major Revision

Reviewers' comments:

Reviewer's Responses to Questions

**Comments to the Author**

1. Is the manuscript technically sound, and do the data support the conclusions?

Reviewer #1: Yes

Reviewer #2: Yes

2. Has the statistical analysis been performed appropriately and rigorously? 

Reviewer #1: Yes

Reviewer #2: No

3. Have the authors made all data underlying the findings in their manuscript fully available?

Reviewer #1: Yes

Reviewer #2: No

4. Is the manuscript presented in an intelligible fashion and written in standard English?

Reviewer #1: Yes

Reviewer #2: Yes

5. Review Comments to the Author

Reviewer #1: Comments to author

The authors have conducted a well-structured study exploring the genetic basis of nutrient acquisition in pearl millet through ionomic profiling and GWAS. The correlation of leaf ion content with root anatomical and agro-morphological traits further strengthens the study’s relevance. The identification of genomic regions and candidate genes linked to leaf ion concentrations offers promising avenues for breeding nutrient-efficient and climate-resilient varieties. Overall, the authors have done commendable work with a sound research design, making the study highly valuable for advancing nutrient-efficient pearl millet breeding. However, I have some minor concerns with the manuscript which are mentioned below:

1. I would suggest considering a modification of the manuscript title to better reflect the key focus of the study.

2. The quality of some figures needs improvement. Please ensure that all figures, including those in the supplementary section, are clear, high-resolution, and appropriately labelled for better clarity and presentation. Additionally, maintain a proper font size for readability in these figures.

3. Ensure all abbreviations must be defined at their first mention throughout the manuscript and in full in captions (all tables, figures and supplementary files). I also recommend a thorough proofreading of the manuscript to correct typographical and grammatical errors.

4. Lines 83-92 should be modified to focus on discussing the importance of the study, the rationale behind it, and the objectives it aims to achieve. Please avoid discussing results in this section.

5. The Methods and Materials (M&M) section is overly detailed and could benefit from a more concise presentation. I encourage the authors to reconsider the level of explanation to better align with the standard format typically expected in research papers and to cite relevant sources wherever the methodology has been previously described.

6. Have the authors used individual data from both years separately and was the data pooled for analysis? Mention clearly. If pooled, did the authors test for homogeneity of error variances across years or a similar approach?

7. Add a table in the supplementary section showing the significance of ion content across different soil depths, locations, and over two years to strengthen the robustness of their findings.

8. Line 96 mentions 160 pearl millet accessions, while line 173 refers to 165 pearl millet accessions. Please verify and correct the discrepancy.

9. Line 245: Please write the equation using the equation editor in Word.

10. Wherever applicable, please include the proper manufacturer or developer details, including links to the instrument, software, or package, in brackets throughout the text.

11. Lines 297 & 326: The authors are advised to properly discuss the PCA results (for both soil and leaf) and heritability estimates (for soil) for both ion content analysis in both the results and discussion sections. This should also include the synergistic and antagonistic roles among ions and later please elaborate on what was achieved by this analysis and justify the findings with supporting studies.

12. Consider moving Table S2 as Table 1 (soil ion) and Table 1 as Table 2 (leaf ion). These can also be merged and presented as single table. Similarly, Fig. 1 and Fig. 2 can be merged.

13. Line 359: check line and figure cited.

14. The discussion section appears weak and should be improved to ensure a logical flow of results, where each finding is discussed sequentially and coherently. Additionally, I recommend the authors to incorporate more recent and relevant references to better support and contextualize their results, thereby enhancing the overall scientific impact of the study.

Reviewer #2: Dear Editor,

The manuscript "Genetic control of the leaf ionome in pearl millet and correlation with root and agromorphological traits." by Princia Nakombo-Gbassault is an original article focusing on ionome profiling of soil and leaf samples of pearl millet for two years. Author used different, correlations and GWAS methods to identify MTA associated with key root and morpho-agronomic traits in the in 160 pearl millet core collection. They tried to identify candidate genes for Mg and K. This study has a great deal of the data. The manuscript is very well written with concise introduction but missing clearly stated objectives. Authors nicely covered scientific and general aspects about pearl millet. The material and methods are clearly described except for a few issues raised below. The results were comprehensively presented, interpreted, and were discussed with implications.

I have the following suggestions that could be helpful to make this manuscript more useful to the reader.

Abstract: Please include the objectives of the experiment in the abstract.

L82: Please state the objectives of the experiment clearly. Clear objectives will improve the flow of the paper and inform the reader what to expect.

L83-89: This content should be included in the results section.

L122: Please clarify why author air dried and measured the remaining shoot mass. The purpose of this step is not clear.

L173: In the plant material section, author stated that 160 lines were planted. However, in the genotypic data section, author stated that leaf samples were obtained from 165 samples. Please explain this discrepancy.

L245: The model did not display correctly in the PDF file. If the journal software settings do not allow author to use the functions, please insert an image of the model.

L252: Author mentioned that he performed ANOVA on the trait data. Please include the ANOVA results as a supplementary table.

Figures: Figure 1B is of poor quality. Please replace it with a high-quality figure. The macro/micronutrient names are not visible in the current figure.

Figure 3: Please use lighter colors or make the coloration numbers darker to highlight the significant correlations in this figure. The current strong colors obscure the correlation values. Additionally, please use consistent legend labels: it is Mx_Nbr in the figure title but Nbr_Mx in the actual figure.

Also, in Fig 3, 16 out of 17 ions did not show any consistent relationship with grain weight. Can we say that breeding for any macro/micronutrients is not worth with yield?

Figure 4: Please use a high-resolution figure. The log values are not visible in Figures A and B.

Figure 5A: Please replace this figure with a high-resolution version.

L326: why author did 19 ion analysis in soil and 17 ions in plant leaves?

L359: Please correct the right supplementary figure number. Fig S4 is for box plot in the supplementary files and that does not match with author’s statement with correlations.

L395-397: Author found that more xylem vessels may reduce the nutrient uptake of Fe, Cu, and P. However, this seems contradictory to what is known about pearl millet. Pearl millet is a highly water-use efficient crop, and if it has fewer xylem vessels, it is unclear how it would maintain its high water-use efficiency and drought tolerance. Please address this apparent contradiction and discuss the implications for pearl millet breeding.

L395: S5 fig: Please use a consistent naming system and order across all figures. One trait is missing in Figure S5-B, and the order is changed.

L403-414: author used four methods for doing GWAS. Was any methods being better than other? Did they got all the MTA using combination of different methods? As per author, MTA reported in the study were found by at least two methods. When they used two methods then both methods found the same SNPs for the studied traits?

L425-426: Author suggested that no colocalization was found between QTLs and ions and that they are working independently. However, in L366-368, author mentioned that many traits are interlinked and have a stable relationship. Please clarify this point.

L437: Author found 74 SNPs for cobalt, which is surprising given that cobalt is present in low concentrations compared to other micronutrients. Please discuss whether the 160 (or 165?) breeding lines used in the study are sufficiently representative of the pearl millet core-collection.

L448-449: Please cite the appropriate research paper(s) for the previous GWAS mentioned.

Author found a QTL for Cadmium on Chr 7 in the result section. Please discuss the importance of this trait for pearl millet breeding.

The authors should provide the study's outcome, such as what genotypes have better performance listed MTAs

6. PLOS authors have the option to publish the peer review history of their article (what does this mean? ). If published, this will include your full peer review and any attached files.

**Do you want your identity to be public for this peer review?** For information about this choice, including consent withdrawal, please see our Privacy Policy .

Reviewer #1: **Yes: ** Amit Rana

Reviewer #2: No

---

## [Author Response · Author response to Decision Letter 0]

17 Apr 2025

Response to the editor and reviewers

Journal Requirements:

R: We modified the manuscript to meet PLoS One’s style.

R: This information was added in the Material and Methods section. We did not collect any material but just conducted field trials in close collaboration with the ISRA.

3. Thank you for stating the following financial disclosure: [P.N.G. was supported by a joint PhD grant from the Institut de Recherche pour le Développement (IRD, France) and the French Embassy in the Central African Republic. We acknowledge support from the Royal Society (Anatomics grant ICA-R1-180356 to MB and 785 NK), the Agropolis Fondation (ValoRoot grant n°2202-002 to LL) as part of the "Investissement d'avenir" (ANR-l0-LABX-0001-0l), under the frame of I-SITE MUSE (ANR-16-IDEX-0006), and the European Plant Phenotyping Network (EPPN2020 project Ionomil n°386).].

R: The modified financial disclosure has been added to the cover letter.

Reviewer #1

The authors have conducted a well-structured study exploring the genetic basis of nutrient acquisition in pearl millet through ionomic profiling and GWAS. The correlation of leaf ion content with root anatomical and agro-morphological traits further strengthens the study’s relevance. The identification of genomic regions and candidate genes linked to leaf ion concentrations offers promising avenues for breeding nutrient-efficient and climate-resilient varieties. Overall, the authors have done commendable work with a sound research design, making the study highly valuable for advancing nutrient-efficient pearl millet breeding.

R: We thank reviewer #1 for his constructive and helpful comments.

However, I have some minor concerns with the manuscript which are mentioned below:

1. I would suggest considering a modification of the manuscript title to better reflect the key focus of the study.

R: We could not find a new title that reflects the content of our manuscript better than the current one. We therefore did not change it.

2. The quality of some figures needs improvement. Please ensure that all figures, including those in the supplementary section, are clear, high-resolution, and appropriately labelled for better clarity and presentation. Additionally, maintain a proper font size for readability in these figures.

R: We modified the figures as requested.

3. Ensure all abbreviations must be defined at their first mention throughout the manuscript and in full in captions (all tables, figures and supplementary files). I also recommend a thorough proofreading of the manuscript to correct typographical and grammatical errors.

R: We edited the manuscript as suggested.

4. Lines 83-92 should be modified to focus on discussing the importance of the study, the rationale behind it, and the objectives it aims to achieve. Please avoid discussing results in this section.

R: We changed the text as requested.

5. The Methods and Materials (M&M) section is overly detailed and could benefit from a more concise presentation. I encourage the authors to reconsider the level of explanation to better align with the standard format typically expected in research papers and to cite relevant sources wherever the methodology has been previously described.

R: We have attempted to reduce the text while retaining all the important information, as we are committed to open science and want our work to be as easily replicable as possible.

6. Have the authors used individual data from both years separately and was the data pooled for analysis? Mention clearly. If pooled, did the authors test for homogeneity of error variances across years or a similar approach?

R: We added this information in the Material and Methods section.

The data were first analyzed separately for each year. Then, the datasets from both years were combined to assess the effect of the year. ANOVA assumptions, including the homogeneity of error variances across years, were tested before conducting subsequent analyses.

7. Add a table in the supplementary section showing the significance of ion content across different soil depths, locations, and over two years to strengthen the robustness of their findings.

R: We made this result more easily accessible in Table 1 in the revised version (this information was in Table S2 in the previous version of the manuscript).

8. Line 96 mentions 160 pearl millet accessions, while line 173 refers to 165 pearl millet accessions. Please verify and correct the discrepancy.

R:We thank Reviewers #1 and #2 for alerting us to this discrepancy. We have revised the text to better explain the material that was used.

9. Line 245: Please write the equation using the equation editor in Word.

R: We re-wrote the equation.

10. Wherever applicable, please include the proper manufacturer or developer details, including links to the instrument, software, or package, in brackets throughout the text.

R: All the scripts are available with all the data on a repository site (https://doi.org/10.23708/DWGEAJ). We tried to make sure all the relevant information needed was included but without making the material and methods section too long (see point 5 above).

11. Lines 297 & 326: The authors are advised to properly discuss the PCA results (for both soil and leaf) and heritability estimates (for soil) for both ion content analysis in both the results and discussion sections. This should also include the synergistic and antagonistic roles among ions and later please elaborate on what was achieved by this analysis and justify the findings with supporting studies.

R: We thank the reviewer for this suggestion. Regarding the heritability estimates for soil, we would like to clarify that no heritability analysis was performed for soil ion content as part of our study, as the data on soil ions do not allow for a robust heritability calculation. However, we did calculate the heritability for ion content in the leaves, and we have added further discussion of these results, along with a more detailed analysis of the PCA for both leaf and soil ion content.

12. Consider moving Table S2 as Table 1 (soil ion) and Table 1 as Table 2 (leaf ion). These can also be merged and presented as single table. Similarly, Fig. 1 and Fig. 2 can be merged.

R: We changed the tables as suggested. Table S2 is now Table 1, and Table 1 is now Table 2. We did not merge these two tables as they represent different results, and merging them would have created a table too large for easy reading.

13. Line 359: check line and figure cited.

R: We have corrected the text.

14. The discussion section appears weak and should be improved to ensure a logical flow of results, where each finding is discussed sequentially and coherently. Additionally, I recommend the authors to incorporate more recent and relevant references to better support and contextualize their results, thereby enhancing the overall scientific impact of the study.

R: We edited the discussion thoroughly to try to address this comment.

Reviewer #2

The manuscript "Genetic control of the leaf ionome in pearl millet and correlation with root and agromorphological traits." by Princia Nakombo-Gbassault is an original article focusing on ionome profiling of soil and leaf samples of pearl millet for two years. Author used different, correlations and GWAS methods to identify MTA associated with key root and morpho-agronomic traits in the in 160 pearl millet core collection. They tried to identify candidate genes for Mg and K. This study has a great deal of the data. The manuscript is very well written with concise introduction but missing clearly stated objectives. Authors nicely covered scientific and general aspects about pearl millet. The material and methods are clearly described except for a few issues raised below. The results were comprehensively presented, interpreted, and were discussed with implications.

R: We thanks reviewer #2 for his/her constructive and helpful feedback.

I have the following suggestions that could be helpful to make this manuscript more useful to the reader.

Abstract: Please include the objectives of the experiment in the abstract.

R: The objective of the study is indicated in the abstract line 21-23.

L82: Please state the objectives of the experiment clearly. Clear objectives will improve the flow of the paper and inform the reader what to expect.

R: We stated the objectives at the end of the introduction as suggested.

L83-89: This content should be included in the results section.

R: We removed this text as suggested.

L122: Please clarify why author air dried and measured the remaining shoot mass. The purpose of this step is not clear.

R: Shoot dry biomass was measured to evaluate the growth of the different genotypes at the time of phenotyping. This information was added in the text.

L173: In the plant material section, author stated that 160 lines were planted. However, in the genotypic data section, author stated that leaf samples were obtained from 165 samples. Please explain this discrepancy.

R: We thank Reviewers #2 for alerting us to this discrepancy. We have revised the text to better explain the material that was used.

L245: The model did not display correctly in the PDF file. If the journal software settings do not allow author to use the functions, please insert an image of the model.

R: We changed the file to make the model available.

L252: Author mentioned that he performed ANOVA on the trait data. Please include the ANOVA results as a supplementary table.

R: The results of the ANOVA are presented in Tables 1 and 2 (revised version). The effect of Year is shown for leaf ions and soil content, while the effect of Depth is shown for soil ion content. Significant values are indicated in bold.

Figures: Figure 1B is of poor quality. Please replace it with a high-quality figure. The macro/micronutrient names are not visible in the current figure.

R: We have made the modifications as requested. Figure 1B has been replaced with a high-quality version, and the names of the macro- and micronutrients are now clearly visible.

Figure 3: Please use lighter colors or make the coloration numbers darker to highlight the significant correlations in this figure. The current strong colors obscure the correlation values. Additionally, please use consistent legend labels: it is Mx_Nbr in the figure title but Nbr_Mx in the actual figure.

R: We have adjusted the figure by using lighter colors and darkening the correlation values to improve visibility. We have also corrected the legend labels to ensure consistency between the figure title and the labels - now all follow the same format.

Also, in Fig 3, 16 out of 17 ions did not show any consistent relationship with grain weight. Can we say that breeding for any macro/micronutrients is not worth with yield?

R: This is an interesting point. The relationship between leaf ion content and yield is complex and dependent on the environment, and we cannot draw any conclusions from the lack of correlation. We have added this information to the text. In our conditions, Mn seems to be a limiting factor, as indicated in the discussion.

Figure 4: Please use a high-resolution figure. The log values are not visible in Figures A and B.

R: We have updated Figures A and B with high-resolution versions as requested.

Figure 5A: Please replace this figure with a high-resolution version.

R: We have updated Figure 5A with high-resolution versions as requested.

L326: why author did 19 ion analysis in soil and 17 ions in plant leaves?

R: Arsenic (As) and chromium (Cr) were also measured in plant leaves, but only in 2022. To maintain consistency and balance in the leaf ion dataset across all years, these two ions were excluded from the final analysis.

L359: Please correct the right supplementary figure number. Fig S4 is for box plot in the supplementary files and that does not match with author’s statement with correlations.

R: We have corrected the supplementary figure numbering to accurately match the content.

L395-397: Author found that more xylem vessels may reduce the nutrient uptake of Fe, Cu, and P. However, this seems contradictory to what is known about pearl millet. Pearl millet is a highly water-use efficient crop, and if it has fewer xylem vessels, it is unclear how it would maintain its high water-use efficiency and drought tolerance. Please address this apparent contradiction and discuss the implications for pearl millet breeding.

R: We thank the reviewer for highlighting this point. In our previous analysis, we did not apply the same genotype filtering criteria to the root anatomical and agromorphological data as we did for the leaf ionome. Specifically, we had included genotypes that did not flower at 49/42 DAS, leading to a mismatch in the genotypes used for leaf ionomics and root/agro traits. We have now corrected this by filtering all datasets to include only the same genotypes, which did not change the overall trends but affected the significance of some results. These changes are reflected in the revised Results section. Regarding L395-397, we acknowledge the contradiction. Pearl millet’s water-use efficiency and drought tolerance are complex traits. We added some discussion on this line 1055-1068.

L395: S5 fig: Please use a consistent naming system and order across all figures. One trait is missing in Figure S5-B, and the order is changed.

R: We have updated Figure S5 to ensure a consistent naming system and order for all traits.

L403-414: author used four methods for doing GWAS. Was any methods being better than other? Did they got all the MTA using combination of different methods? As per author, MTA reported in the study were found by at least two methods. When they used two methods then both methods found the same SNPs for the studied traits?

R: GWAS corrects for association with population structure and/or relatedness through an estimated kinship matrix, and traits might more or less co-vary with both. Having models that make slightly different corrections and assumptions helps increase the robustness of the results. By applying multiple methods and retaining only the MTAs identified by at least two of them, we aimed to increase the robustness of the results. We always used four methods and only retained the SNPs that were detected by at least two methods. In that case, both methods identified the same SNP.

L425-426: Author suggested that no colocalization was found between QTLs and ions and that they are working independently. However, in L366-368, author mentioned that many traits are interlinked and have a stable relationship. Please clarify this point.

R: This is a very good point. Correlations in concentration could be explained by different mechanisms, such as co-localization in the soil profile or common transport mechanisms, for example. We have added some text to clarify this point (lines 1074-1079)

L437: Author found 74 SNPs for cobalt, which is surprising given that cobalt is present in low concentrations compared t

---

## [Editor Report · Decision Letter 1]

23 Apr 2025

Genetic control of the leaf ionome in pearl millet and correlation with root and agromorphological traits

PONE-D-25-04942R1

Dear Dr. Laplaze,

We’re pleased to inform you that your manuscript has been judged scientifically suitable for publication and will be formally accepted for publication once it meets all outstanding technical requirements.

Kind regards,

Prasanta K. Subudhi, Ph.D.

Academic Editor

PLOS ONE

Additional Editor Comments (optional):

The authors revised the manuscript addressing the concerns of the reviewers. The manuscript may now be accepted for publication.
---

## [Editor Report · Acceptance letter]

PONE-D-25-04942R1

PLOS ONE

Dear Dr. Laplaze,

I'm pleased to inform you that your manuscript has been deemed suitable for publication in PLOS ONE. Congratulations! Your manuscript is now being handed over to our production team.

Kind regards,

on behalf of

Dr. Prasanta K. Subudhi

Academic Editor

PLOS ONE